# Stranded Assets as a Key Concept to Guide Investment Strategies for Sustainable Development Goal 6

**Robert M. Kalin [1,\*], Joseph Mwanamveka [2], Andrea B. Coulson [3] , Donald J. C. Robertson [1], Holly Clark [1], Jon Rathjen [4] and Michael O. Rivett [1]**

[1] Climate Justice Fund Water Futures Programme, Department of Civil and Environmental Engineering, University of Strathclyde, Glasgow G1 1XN, UK; donald.j.robertson@strath.ac.uk (D.J.C.R.); holly.clark@strath.ac.uk (H.C.); michael.rivett@strath.ac.uk (M.O.R.)

[2] Minister for Agriculture, Irrigation and Water Development, Government of Malawi, Lilongwe, Malawi; info@cjfwaterfuturesprogramme.com

[3] Department of Accounting and Finance, University of Strathclyde, Glasgow G1 1XN, UK; a.b.coulson@strath.ac.uk

[4] Water Industry Team, Scottish Government, Victoria Quay, Edinburgh EH6 6QQ, UK; jon.rathjen@gov.scot

\* Correspondence: Robert.Kalin@strath.ac.uk

**Abstract:** The concept of Stranded Assets has been used for nearly 50 years across many sectors, most recently it has been a focus of investment portfolios in light of the possible impacts of climate change. However, to date there has been no in-depth determination of the impact of Stranded Assets for rural water supply, despite international development targets from Rio, through Millennium Development Goals (MDGs), and now the Sustainable Development Goals (SDGs). The limiting factor for carrying out such an assessment is the requirement of a full and detailed asset register recording all rural water supplies in a country. The Scottish Government Climate Justice Fund Water Futures Programme, in collaboration with the Government of Malawi, is undertaking a comprehensive asset audit across Malawi, and this paper introduces the concept of Stranded Assets for the rural water supply sector using Malawi as an exemplar. Here, we demonstrate how significant change in the implementation strategy for SDGs compared to the MDGs is needed to reduce the potential for Stranded Assets and meet its ultimate aim.

**Keywords:** Stranded Assets; rural water supply; Sustainable Development Goals; WASH; Malawi

## 1. Introduction

Considerable research efforts have been placed on the symptoms of failure around rural water supply and enhanced service provision investments in Africa and Asia. These investments include community supply boreholes, solar groundwater supplies, low-cost or gravity fed piped water networks, as well as major irrigation and water supply investments [1–7]. From an investment portfolio perspective, it is strange the sector has not addressed the devaluation, loss of return, and resulting liabilities that result from poorly implemented assets. The concept, if not the terminology, of Stranded Assets is well known and has been embedded in investment planning for nearly 50 years or more [8,9]. This includes use in the water sector, for example the potential stranding of irrigation infrastructure from drought management in the Murray–Darling Basin, Australia [10,11]. Despite this, the rural water supply sector failed to achieve global sustainability starting in the 1980s with the Rio summit, falling short through the Millennium Development Goal (MDG) period to 2015, and is struggling in the Sustainable Develop Goal (SDG) era to 2030. The sector has sadly not yet grasped the

important investment behaviours guided by the concept of Stranded Assets as a driver to meet the challenge of a sustainable planet.

What is a Stranded Asset? There are many definitions of a Stranded Asset depending on (a) the sector (energy, utilities, etc.), or (b) the challenge (climate change, global markets), or (c) the discipline (accounting, economics, engineering) [9,10,12,13]. For the purpose of this paper we define a Stranded Asset as '*Water Sector infrastructure assets that have prematurely lost financial value or are devalued before the end of design lifetime, assets still within design lifetime but due to improper policy or management do not provide the intended service provision and/or have been abandoned, and assets that are converted to a social, environmental or financial liability before the end of design lifetime*'. Stranded Assets arise when physically, it does not provide service; socially, where a community does not use a functioning resource (i.e., it was built in a socially inappropriate way); or economically, where the cost of delivering the service is substantially greater than the net value of the 6resource. Planning for SDG number 6 (ensuring water and sanitation for all) needs to consider all future investments against long-term economic development strategies.

Reducing the number of Stranded Assets requires robust analysis and sound planning within a supportive national policy framework. If we are ever to achieve sustainability targets, financial instruments for the water sector, including private donors and service organizations, must consider this concept in detail when planning investments. This is particularly the case for SDG6 [14] and ultimately beyond 2050 because, unlike the MDGs, the SDGs focus on household/individual access to water services. Up to date information on asset status and review of the asset implementation framework must be undertaken on a regular basis by those providing and managing capital investment. It is noteworthy that even the recent Organization for Economic Co-operation and Development (OECD) guidance for sustainable investment in the water sector has failed to recognize the importance of Stranded Assets [15]. Stranded Assets impacts are felt in many ways, but ultimately result in reduced economic growth potential, reduction in innovation (as many Stranded Assets are failed innovations (by non-governmental organizations (NGOs) that have not been decommissioned), and, as highlighted later, pose a risk to systemic sustainability and the health and wellbeing of the communities who are dependent on these assets.

Analysis of Stranded Assets in the rural water sector requires national scale evaluation by the State, water governing body, and where relevant, public or private providers. This paper will reference recent efforts to develop a national Management Information System (MIS) for rural water supplies in Malawi, undertaken jointly by the Scottish Government's Climate Justice Fund Water Futures Programme (CJF), implemented by the University of Strathclyde together with the Government of Malawi. Through programme findings, it became apparent that Malawi's water sector faces a considerable and multi-faceted challenge with Stranded Assets [16–24].

Since 2011 the CJF program has been evaluating rural water supply infrastructure installed in Malawi by NGOs and donors and found that many assets do not meet design specifications, do not deliver the service provision within the time period of the design, and fail to provide the service level they planned to deliver. Stranded Assets within rural water supplies have become a social and economic liability to the Government of Malawi. Given the majority of rural water supply assets were planned and implemented independently of the Government of Malawi, it will be important that the entire water sector (implementing partners and donors) works together to recognize and respond to what can be reasonably considered to be a shared accountability for current water resource liabilities.

In particular, our case study highlights the Water Resources Act Malawi, and the 2018 regulatory framework for the Malawi National Water Resources Authority (NWRA) provided an opportunity to address Stranded Assets through adoption of updated water policy and an active governance framework. Reliance was then on the non-governmental sector to ensure rigorous adoption of government standards, and ultimately to reduce the legacy of Stranded Assets and reduce the risk of creating new Stranded Assets. We evaluate the success of this governance structure in light of Stranded Assets documented to date and SDG6 targets proposing a way forward.

## 2. Materials and Methods

### 2.1. Study Area

Malawi is located in Southeastern Africa, and shares borders with Mozambique, Zambia, and Tanzania (Figure 1). The current population of Malawi is 17.4 million (2018 Census Data). During the MDG era, Malawi has undertaken economic and structural reforms to enhance its economic growth. Though extreme poverty has declined, poverty remains widespread, with the national poverty rate increasing slightly from 50.7% in 2010 to 51.5% in 2016, mainly driven by population growth. Malawi's energy sector is heavily dependent on hydropower and Malawi's GDP is strongly linked to the successes and failures around rural small holder agricultural performance. Eighty percent of the population of Malawi live in rural areas and are dependent upon groundwater, therefore sustainability of rural water supplies and national water resource management is a threshold factor that underpins long-term economically stability.

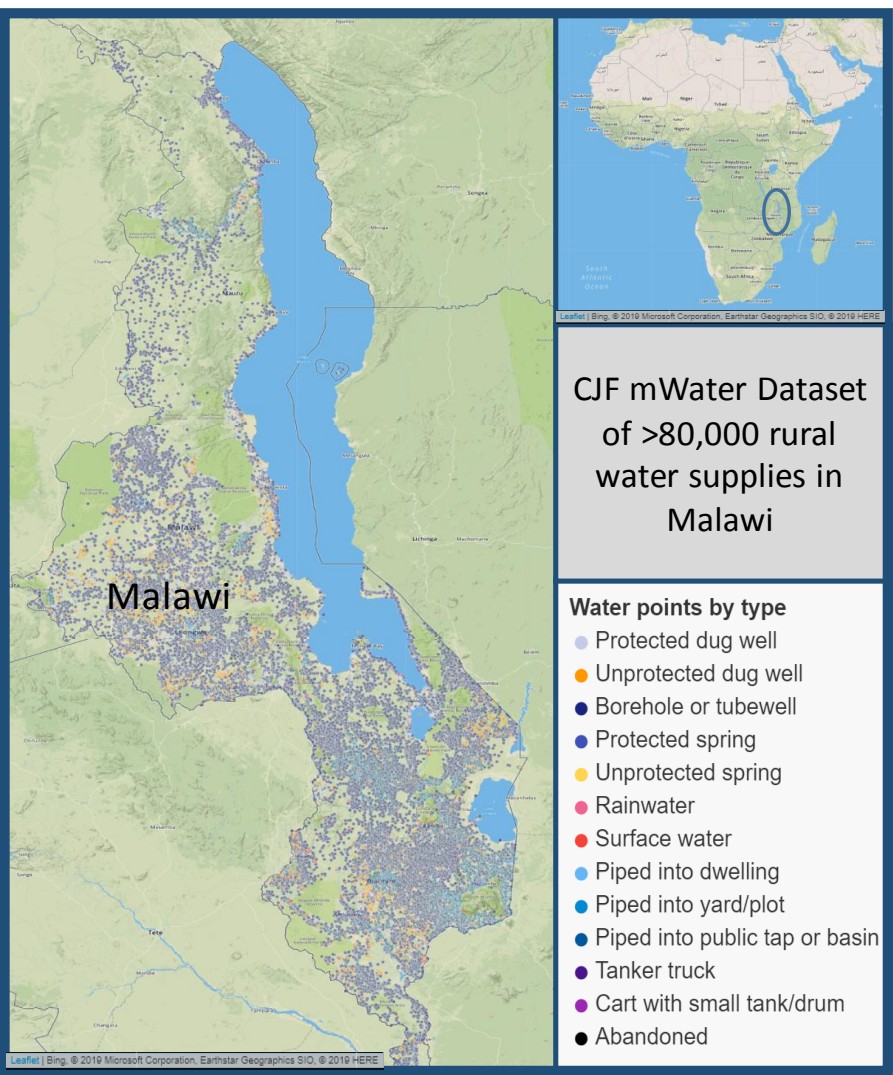

**Figure 1.** Map of Malawi with locations of currently known rural water supply assets (as of 1 March 2019).

### 2.2. Climate Justice Fund Water Future Programme Approach

Given the importance of the rural population base in Malawi, the CJF Programme—in partnership with the Government of Malawi—is developing an MIS for the rural water sector in Malawi [21]. The goal of the MIS is to provide a basis for long-term strategic management for the WASH sector

infrastructure in Malawi. The MIS, hosted on the mWater platform (www.mwater.co), currently contains millions of pieces of data for hundreds of thousands of water and sanitation facilities across Malawi. The CJF programme is providing funding to visit every rural water point and evaluate the holistic sustainability of each on a point by point basis. This also includes evaluation of all the local sources of contamination risk to each water supply (pit latrines and waste sites). These data are being collected by Government of Malawi staff and are uploaded live, with quality control checks in place, into the MIS giving live access for all District, Regional, and National Governments of Malawi staff to support information-led decision making. As the national asset dataset is completed in 2019, the MIS offers a significant base for developing and guiding investment in the rural water and sanitation sector nationally, and importantly to evaluate the root causes of Stranded Assets. Further detail on the design and development of the mWater MIS is provided by Miller et al. [21].

As of 1 March 2019, there is up to date information within the mWater MIS on nearly 57,000 of over 80,000 improved rural water supplies the CJF programme projects to exist across Malawi. We draw upon this current dataset that fully covers Southern Malawi, much of Central Malawi, but only limited parts of Northern Malawi—the last region where we are currently collecting data as a basis for our identification of Stranded Assets.

## 3. Results and Discussion

### 3.1. Assessing Stranded Assets at a National Level

Figure 2 presents summary data for over 55,000 rural water point functionality surveys (borehole, gravity fed system, etc.), disaggregated into 4 distinct classes described below. The percentage occurrence as a proportion of all rural water points in the dataset is indicated below in parentheses:

1.  Functional (52.9%)—where the water resource asset is functioning as designed and providing improved water service to the community as designed;
2.  Partially Functional, i.e., functional but with problems (21.6%)—where the water resource asset provides water intermittently as a result of a range of issues such as:

    -   Poorly installed water point affected by decline in groundwater table resulting in a dry water point during some months (Stranded Asset),
    -   Poorly installed water point or low aquifer yield resulting in a water point running dry on a daily basis (Stranded Asset),
    -   Poorly maintained water point or water system resulting in limited access to water throughout the year (Stranded Asset),
    -   Poorly installed water point into a water resource that is contaminated or has been contaminated (e.g., salinity and co-location of pit latrines and waste) (Stranded Asset),
    -   Poorly managed water point (issues with tariff setting/collection, non-professional management, mis-management of resources, lack of capacity) (Stranded Asset);

3.  Non-Functional (22.3%)—where the water resource asset does not supply water (Stranded Asset);
4.  No Longer Exists or Abandoned (3.2%)—where the water resource asset has been fully abandoned (Stranded Asset).

The CJF water point asset assessment data is further ground-truthed through the Programme's Forensic Analysis of Stranded Assets [19], undertaken to date on 1% of rural water points that fall into classes 2(a–e), 3, and 4 in Figure 2 above. This work is undertaken in order to elucidate the scale of the problem (socially and economically) [23] and the Root Cause of Failure that brought about the devaluation of the use-value of these assets.

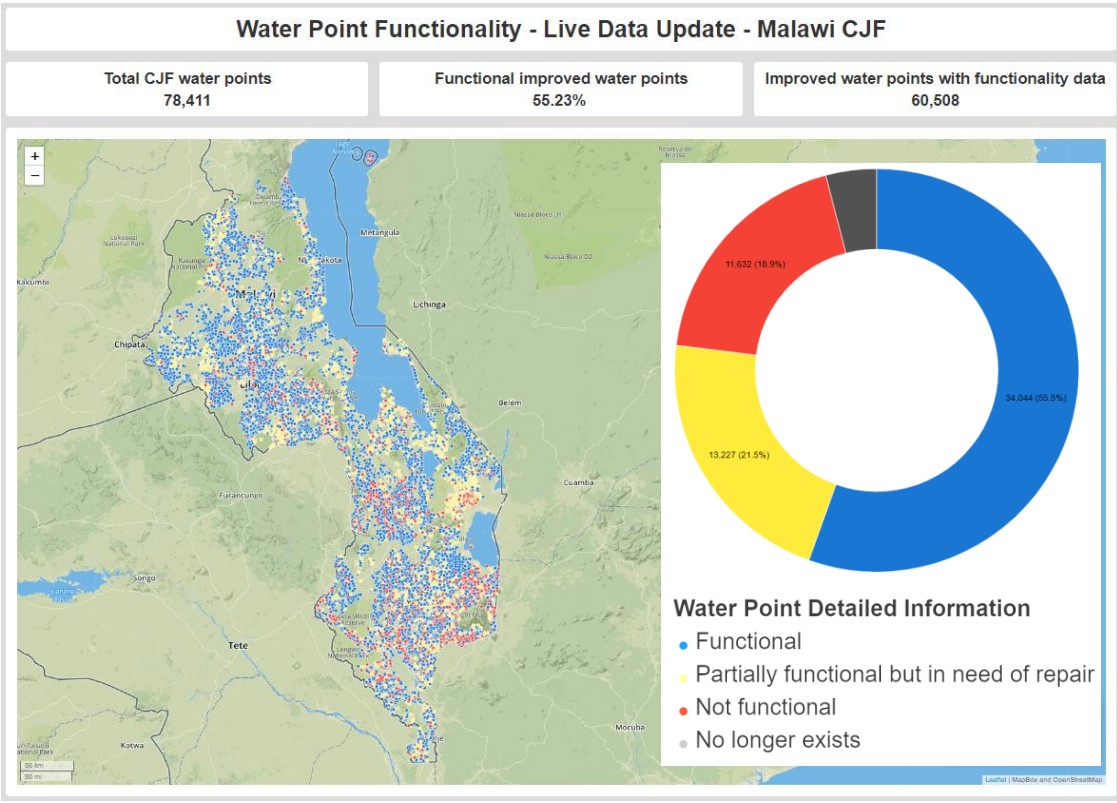

**Figure 2.** Results of site surveys to evaluate functional status of rural water supplies as presented by the mWater Management Information System (MIS).

Figure 3 depicts an example of a Stranded Asset determined through the MIS of the CJF Programme. Here, a borehole was not installed to Government of Malawi Standards. The Afridev hand-pump is the Gov't of Malawi standard as installed at this site. The Government standards require a pumping test of the drilled borehole before installation of the Afridev hand-pump. The required flow rate of the borehole should provide 0.25 L/s for 4 h as the Afridev hand-pump provide 0.33 L of water with each stroke and therefore has a maximum flow rate of 0.33 L/s if pumped continually. During the asset assessment, each water point is tested for flow rate and if it is not capable of delivering water at this flow rate year-round (except during minor repairs)—it is a Stranded Asset (i.e., it does not deliver to the design criteria resulting in a loss of value). The example in Figure 2 is a rural supply borehole in Chikwawa District, Malawi that was commissioned by an NGO, but can only provide water at 0.25 L/s for 2.25 min of initial pumping after which the community must wait hours per day for water levels to recover. This water supply never provided sufficient water to the community and therefore is not only a stranded physical asset that has lost its economic usefulness and therefore financial value, but as the community health and well-being is affected, it is a Stranded Asset in social terms, posing a risk to what may be termed the social capital of the community [25,26]. It should never have been commissioned and should be decommissioned per Government of Malawi statutes (filled with cement bottom to top and the site fully returned to original status). It is now a Stranded Asset classed as Non-Functional within the National MIS register.

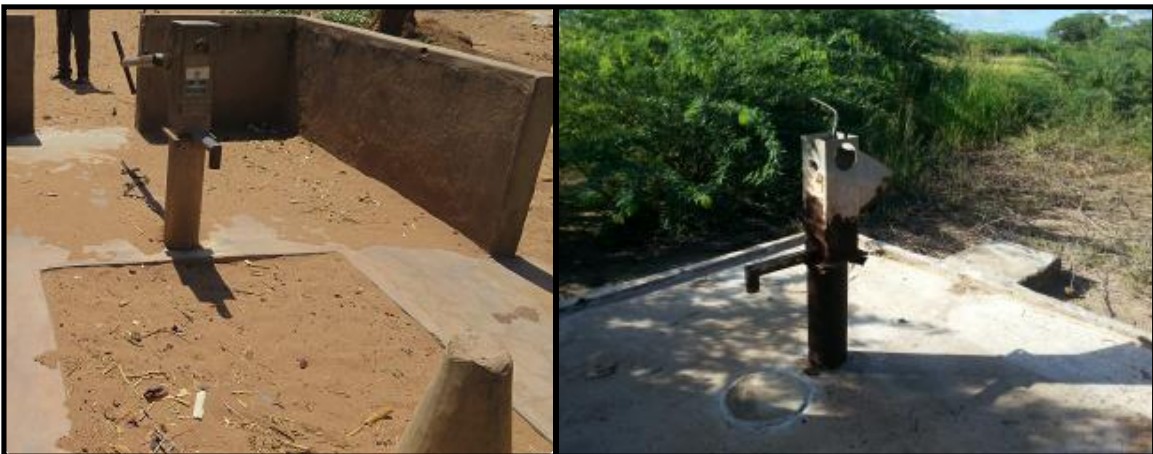

**Figure 3.** Photograph of a rural water supply that has been deemed a Stranded Asset due to poor borehole yield (**left**) and a rural water supply stranded due to poor water quality (**right**).

Similarly, according to Government of Malawi Standard Operating Procedures (SOPs), the water quality must be tested during commissioning (before the pump is installed) and if the water quality falls below Government of Malawi standards, the borehole must be properly decommissioned (filled with cement and the site returned to original status). The other rural water point shown in Figure 3 was completed and commissioned by an NGO, and provides water at appropriate flow rate, however the electrical conductivity (measure of salinity) for this water point during forensic analysis was 14,240 μS/cm. The Government of Malawi Standard allows for a maximum of 3500 μS/cm for use as potable water, and the measured value (expressed as TDS—total dissolved solids) is 3 times over the 1000 mg/L TDS increasingly unpalatable transition point recognized by the WHO. Clearly, this will have a long-term impact on the health and well-being of the community dependent on this water supply, and there is a clear need to reconsider investment planning in areas of highly saline groundwater [24]. The vulnerability of borehole assets to other water-quality concerns such as geogenic sources of fluoride and arsenic, and increasing possibilities of anthropogenic inputs (from sanitation, agricultural, or urban land uses) is often poorly known; routine monitoring is often limited with a significant dependence on research studies [16,20,22,24,27–30].

It is the financial and contractual obligation of the implementing organization to plan and design the investment and to follow existing standards. Those who are paying for the infrastructure have the moral and legal responsibility to make sure works are undertaken and supervised to these standards. It is not possible to transfer responsibilities to the contractor through inappropriate contractual arrangements. However, we have found evidence through our research in the CJF of implementing organizations that fail to plan for and manage the risks (and financial liabilities) associated with Water Resource Investments [31], which are inherently uncertain and never 100% successful. Unfortunately, this reality is generally ignored by much of the rural water sector in Malawi. The result is an ever-increasing financial liability across Malawi due to Stranded Assets.

*3.2. Stranded Assets and SDG6*

The year 2015 saw the end of the Millennium Development Goals (MDGs) and the ushering in of the 2030 agenda. The SDGs, with its 17 individual goals, set out ambitious targets for the international community. SDG6, ensuring access to sustainably managed water and sanitation services for all, departs from MDG7 in expectations and elevates the threshold for success. Thus, driven by the 2030 agenda, investment plans and strategies must recognize SDG6 and its specific targets. In terms of drinking water and sanitation service delivery, where the majority of Stranded Assets are found in Malawi, they are now governed under SDG6.1 and SDG6.2. Table 1 sets out a side-by-side comparison of MDG and SDG targets and indicators. Adding to this evolutionary drive to recognize Stranded

Assets and manage the risk of these is the impact of climate change, further accelerating the need for evolution and action.

**Table 1.** The evolution from the Millennium Development Goals (MDG) to the Sustainable Develop Goals (SDG) sees an increase in expectations. These expectations are set out in the SDG6 targets and indicators.

| MDGs | | SDGs | |
|---|---|---|---|
| Targets | Indicators | Targets | Indicators |
| 7C: Halve the proportion of people without sustainable access to safe drinking water and basic sanitation by 2015 | 7.8: Proportion of population using an improved drinking water source | 6.1 By 2030 achieve universal and equitable access to safe and affordable drinking water for all | 6.1.1 Proportion of population using safely managed drinking water services |
| | Proportion of population using an improved sanitation facility | 6.2 By 2030 achieve access to adequate and equitable sanitation and hygiene for all, and end open defecation paying special attention to the needs of women and girls and those in vulnerable situations | 6.2.1 Proportion of population using safely managed sanitation services, include a hand-washing facility with soap and water |

For the provision of drinking water, as set out in SDG6.1, emphasis can no longer solely be placed on the source of water provision but must be placed on the service level being delivered to the household. It is reasonable to accept this water supply provision should be designed and managed with a lifetime in decades. The focus is shifting away from individual access to a sustainable 'improved source' (as set out in the MDGs) towards access to a 'safely managed' water service, encompassing not only availability, but also accessibility and water quality (Figure 4). What is therefore clear is that in order to achieve progress towards SDG6.1, a planned shift away from the status quo of donor-led community-based water provision to policy-led investment is required.

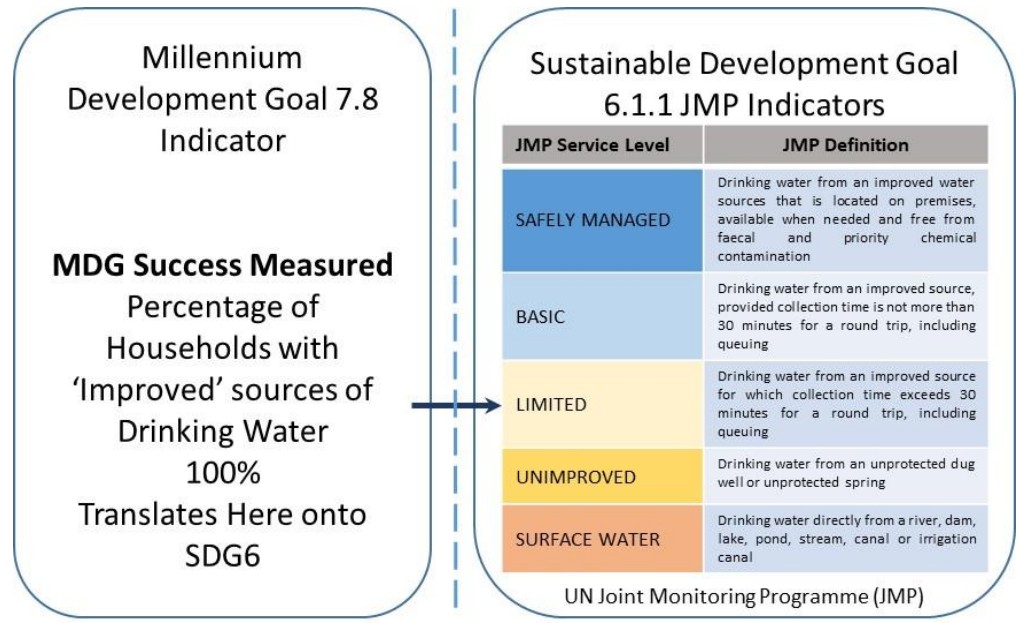

**Figure 4.** The evolution of targets from the MDGs to the SDGs results in the current status (Improved) now being either Unimproved or Limited Service Levels.

However, detailed investment planning supported by data is needed to move from communal water supply initiatives such as hand pumps, favoured by many donors and NGO's, to enhanced service provision with operational life-spans of 50 to 80 years or more. Starting the process

demonstrates ambition towards attainment of SDG6. Crucially, to address SDG6 'Safely Managed' requires investment strategies for professionally managed water and wastewater treatment modalities (moving away from incumbent community-based management models (CBM), and provision of water available on a household's premises that is free from contamination (Figure 5). Therefore, the concept of Stranded Assets plays a vital role in assuring the longevity of investments.

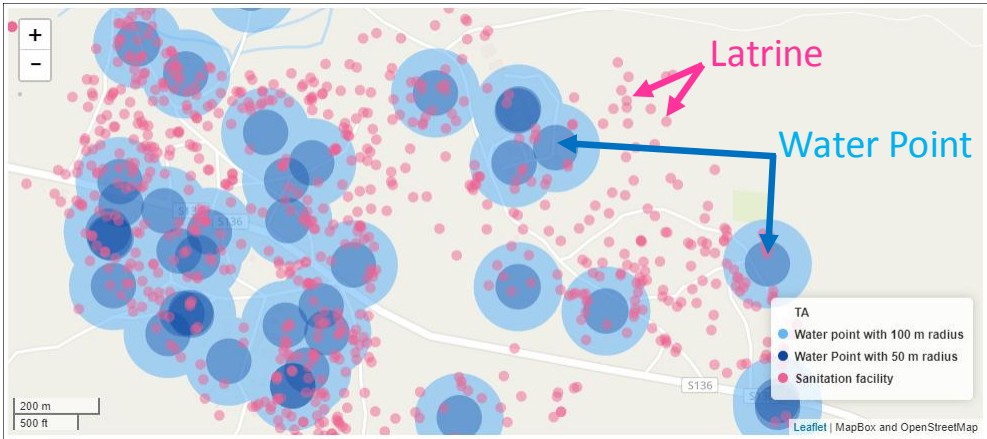

**Figure 5.** Co-location of pit latrines and rural water supplies has increased the risk of contamination and disease, resulting in an expectation that both the water resource (groundwater) and water point (borehole) will have reduced life-spans due to major contamination of groundwater (from the Climate Justice Fund Water Futures Programme (CJF) mWater data console).

In a similar manner to water provision, the SDG sanitation and hygiene targets (through their respective indicator 6.2.1) have been set on the basis of service provision. The past two decades have rightly focused on reducing morbidity and enhancing health and wellbeing, therefore the advances made should not be discounted, but what was not considered during the MDG implementation period was the combined impact of separate and distinct water and sanitation targets.

The MIS has flagged up an increasing proliferation of hand-dug unimproved pit latrines co-located with potable water supplies (hand pumps) in each community (Figure 5). Whilst there is some contradictory Malawian guidance on minimum separation distances between latrines and water points, at 30–50 m spacing [16], it is unclear if such spacing will offer adequate protection in all environments and, moreover, such guidance is not always implemented—around 20% of the cases in a Southern Malawi valley study reviewed by Back et al. [16] were less than 30 m spacing. Furthermore, unimproved pit latrines have been designed to be abandoned after use (98.6% of the nearly 160,000 points surveyed are to be abandoned) creating a new point source of pollution and a long-term community burden. There is an urgent need to begin the planning process that will invest in long-term (50+ year) solutions to sanitation and hygiene which can be environmentally and economically regulated to meet sustainability targets and protect water resources. The current practice, if not changed, will result in a long-term environmental burden on the shallow groundwater supplies, resulting in abandonment of water points due to contamination (Stranded Assets). A clear regulatory framework is needed to address this practice and ultimately reverse the increasing burden of contamination sources and long-term risks to human health.

The result from Malawi shows that what were acceptable development practices under the MDGs need to be challenged, and new approaches led by government regulation and policy are vital to translate acceptable practice and implementation that address the SDGs. It is clear that the continuation of pre-2015 approaches to WASH development will fail to address the challenge posed in achieving the Sustainable Development Goals, ultimately to ever increasing social, economic, and environmental liabilities as Stranded Assets proliferate.

*3.3. Information Supporting Sustainability Targets*

In 2018, the CJF Programme recognized the challenges of SDG6.1 and SDG6.2, given these metrics or indicators require benchmarking of assets to the individual household level. Therefore, during discussions in July 2018 the Scottish Government and Malawi Government agreed to develop a pilot decision support tool within the mWater MIS platform that can both strictly track the UN Joint Monitoring Programme (JMP) indicators [32] (measured at nearly 8500 households), and provide an initial framework at the community level that guides investment, recovers and reduces Stranded Assets, and maximizes positive impact on the health and well-being of the communities. This is achieved through widening the forensic analysis of Stranded Assets, while at the same time bringing them into full use and/or targeting new sustainable asset investment to the locations of highest need and impact for specific communities and households. This 'CJF Investment Approach' captures information required to report against SDG6.1 and SDG6.2 indicators and automatically calculates service levels within the mWater data visualization console, to combine existing and new data from the forensic analysis and benchmarking of local assets.

Table 2 presents results from one of the communities (group village) in Chiradzulu, Zomba and Mangochi Districts in Malawi, where the SDG6.1 household indicator was determined before and after the forensic analysis and targeted rehabilitation of water points that were designated as partially functional (Stranded Assets) within the national MIS dataset. Before intervention, only 8% of the households (n = 127) had a 'Basic' level of service (defined as drinking water from an improved source free of contamination, provided the collection time is not more than 30 min for a round trip including queuing time). After identification of the root cause of the Stranded Assets and subsequent rehabilitation, 61% of the households reported a 'Basic' level of service, demonstrating the potential of the CJF Approach. It should be noted the Government of Malawi has yet to set the national indicators and targets for SDG6 and as such these data are based solely on the JMP SDG6.1 Indicators.

**Table 2.** SDG6.1 Household Survey before and after intervention to rehabilitate Stranded Assets identified in Chiradzulu District Malawi. Indicators were explicitly based on the Joint Monitoring Programme (JMP) definitions as described in Figure 4.

| SDG6.1 Household Survey Group Village Community in Chiradzulu District, Malawi | Insufficient Data from Household to determine Service Level SDG6.1 Indicator | Number of Households with Unimproved or Limited Service Level SDG6.1 Indicator | Number of Households with Basic Service Level SDG6.1 Indicator |
|---|---|---|---|
| Before Rehabilitation of Stranded Assets | 28 (22%) | 89 (70%) | 10 (8%) |
| After Rehabilitation of Stranded Assets | 16 (13%) | 33 (26%) | 78 (61%) |

The transition from donor driven targets/interventions within the sector to aligned proactive planning and strategic investment requires reliable data and coherent public policy. Using the live MIS asset status data in combination with the CJF decision support tool provides a new opportunity to address the challenges of Stranded Assets. In Malawi, it will be important to embrace the opportunities that arise from the recent announcement of the National Water Resources Authority's (NWRA) board (7 November 2018). A strong and collaborative regulatory framework, in combination with a data empowered government, provides an excellent platform on which the sector can build upon.

*3.4. External Risks Contributing to Stranded Assets*

SDG6 indicators and targets should be set by each nation, and therefore no one solution will serve the global aspiration. But there remain consistent challenges, such as poor infrastructure planning, Stranded Assets from poor delivery in the past, and emerging risks such as climate change. Proliferation of rural water supplies as drilled boreholes with Afridev hand-pumps in Malawi means the current investment plans for the rural water supply sector remains heavily dependent on

groundwater resources (and their sustainable management) in Malawi. However, the understanding of groundwater resources [33,34] and the long-term impacts of climate change in Malawi [35–37] is still in its infancy. Investment planning is further limited by assessment of long-term changes in aquifer storage, an absence of accurate drilling records, and a lack of adherence to professional drilling standards and questionable contractual arrangements (e.g., non-payment for a borehole that does not provide adequate water) [19,24]. With the rush to meet MDG targets, the rate of implementation has accelerated in Malawi [23]. Table 3 presents data that shows a large number of shallower installations are Stranded Assets as they do not provide a sustainable water supply throughout the year. Climate change and poor catchment management practices may further impact on groundwater storage resulting in additional Stranded Assets through time.

**Table 3.** Percentage of Stranded Assets where water is only seasonally available from the improved rural water supply as a function of supply borehole depth.

| Depth (m) | Percentage of Water Supplies (%) | | | | | | |
|---|---|---|---|---|---|---|---|
| | 0–10 | 10–20 | 20–30 | 30–40 | 40–50 | 50–60 | >60 |
| Water available year-round | 63.4 | 66.0 | 83.9 | 90.8 | 93.4 | 93.7 | 94.7 |
| Water available only seasonally | 34.5 | 32.4 | 15.0 | 8.8 | 5.8 | 5.5 | 4.1 |
| No information available | 2.1 | 1.6 | 1.1 | 0.4 | 0.8 | 0.8 | 1.2 |
| n = | 1311 | 2288 | 1396 | 4139 | 10,380 | 1745 | 1026 |

When planning investments and reviewing Stranded Assets, the ensuing devaluation in communal hand pump-fitted boreholes as a result of the SDG targets should be seen as a natural step in progression towards sustainable development and toward climate change resilience measures. These distributed water points, which number in the tens of thousands across Malawi, do not serve to progress the 2030 agenda, and with an anticipated lifespan of approximately 25 years per borehole, requires rolling investment plans that must be implemented and operational by 2030. Also, evolution towards a service delivery approach, mandated by SDG6, must be facilitated through the use of appropriate technologies. These initiatives, piped water schemes or equivalent solutions, require greater unification of approach within the sector (led by Government Regulation and Planning) in order to realize the funding and implementation needed. Additionally, loan agreements around Water Infrastructure need to be evaluated carefully by an Economic Regulator. As with the recent OECD guidance [15] this should consider whole of life planning and pay-back periods of 50 years as a minimum (life time of a well-planned Enhanced Service Provision). Economic regulations allow a coherent cost–benefit analysis, to demonstrate the inherent value of investing in centralized piped schemes over continuing the currently dysfunctional distributed approach.

Water services that address climate change adaptation, by way of an integrated water supply system with built-in resilience, provides a much more attractive investment opportunity for the long term. For example, there is already an acknowledged challenge in Malawi with falling groundwater levels and dry-season shortages, both for surface and groundwater. Using the previous disconnected hand pump system, each individual water point is vulnerable to breakdowns, theft, supply failure (e.g., fall in groundwater level), and contamination. However, by adopting a more integrated approach to recognizing potentially Stranded Assets and those at risk, we may begin to address these individual vulnerabilities, not only in adapting to SDG6, but also to the challenges of climate change.

## 4. Recommendations

If we are going to use Stranded Assets globally to guide investment in an SDG6 strategy, we need to recognize:

1.  There must be a robust method to identify the physical and financial nature of Stranded Assets.
2.  There is a need for a robust decision support tool appropriate to lower income countries that can be used to guide point by point assessment of water supply assets.

3.  In terms of a decision support around identified Stranded Assets, this decision needs to be taken based on physical and financial return and clearly indicate if it is worthwhile to 'fix' these or decommission them fully and invest instead in new services.

4.  Inherent within this decision should be considerations of current value in use and financial return on investments alongside considerations of local social circumstances and social values.

5.  SDG6 investment planning of new assets will need to be much more focused on whole of life value linked to long-term returns on investments.

**Author Contributions:** Conceptualization, R.M.K., D.J.C.R., H.C., and J.R.; formal analysis, R.M.K., J.M., A.B.C., and M.O.R.; funding acquisition, R.M.K. and J.R.; investigation, R.M.K., D.J.C.R., and H.C.; methodology, R.M.K. and A.B.C.; project administration, R.M.K. and J.M.; validation, J.R.; writing—original draft, R.M.K.; writing—review and editing, M.O.R.

**Funding:** This research was funded by research by the Scottish Government under the Scottish Government Climate Justice Fund Water Futures Programme research grant HN-CJF-03 awarded to the University of Strathclyde (R.M. Kalin).

**Acknowledgments:** Brewgooder and the Rotary Club Ayr are acknowledged for their efforts in raising funds to financially support the rehabilitation of Stranded Asset community borehole supplies as part of this study.

**Conflicts of Interest:** The authors declare no conflict of interest.

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
