# Peer review of "Stranded Assets as a Key Concept to Guide Investment Strategies for Sustainable Development Goal 6"

_water, doi:10.3390/w11040702_

Round 1

Reviewer 1 Report

The paper looks fine. few comments for improvement:

- Include  Malawi in the keywords,

The introduction needs mor details about water in Africa and Malawi. You may use theses new references:  for drinking water in Malawi: https://www.mdpi.com/1660-4601/16/6/951/htm ; and  for SDG6 in Africa https://www.mdpi.com/2071-1050/11/3/916 

Best wishes

Author Response

The paper looks fine. few comments for improvement:

- Include  Malawi in the keywords,

Response: Done

The introduction needs mor details about water in Africa and Malawi. You may use theses new references:  for drinking water in Malawi: https://www.mdpi.com/1660-4601/16/6/951/htm ; and  for SDG6 in Africa https://www.mdpi.com/2071-1050/11/3/916 

Response: We thank the Reviewer for these insightful publications.    The Article on Water Services Sustainability: Institutional Arrangements and Shared Responsibilities does indeed merit referencing here.    The Bottled Water Industry in Malawi is based on either (a) water supplied by the Water Boards (Companies) or (b) by Companies using shock distillation and neither relate to overall Water Resources Management and we therefore find it difficult to link this publication with the background materials in the introduction (based mainly on water resources access). 

We therefore thank the Reviewer for noting this 2019 publication and we have replaced a cited 2008 publication as this publication is more relevant and more up to date.

Reviewer 2 Report

The paper aims to provide an in-depth 20 determination of the impact of Stranded Assets for rural water supply despite international development targets, from MDGs to SDGs. In addition to this theoretical approach, a case study from Malawi is provided. The paper fits well with the aim of the journal and is well structured. However, few but important suggestions shouls be taken into account in order to improve it:

L46-48: Is the first time in which MDGs and SDGs are cited, please provide a minimum context and aim about both concepts and approaches.

L53: References [Caldwell 2017] and [Roper et al., 2006] should be numbered.

L54-58: If the definition is presented as a citation from an author or official document as appeared between quotation marks (""), pelase provide the source of this definition.

L62: The SDG6 is cited for the first time. Please provide the context and the definition (ensure availability and sustainable management of water and sanitation for all). 

L72: OECD appears for the first time, please do not use the acronym.

L76: NGOs appears for the first time, please do not use the acronym.

L85: Transform the "we recognized" into more impersonal writing.

L89: NGO is used without acronym although it is the second time that this concept appears.

L101: Stranded assets not in capital letters. Apply to all the text of the paper.

L124-125: Figure 1. The Legend is not clearly comprehensible in the map on the left. The location map on the right is too big. Please, reorganize both maps in order to clarify the location map of the stranded assets.

L130 and L145: [Miller et al., 2018] must be replaced by the reference number.

L155: "Figure 1" must be renumbered as Figure 2.

L164-179: Delete "Stranded asset" from the end of all sentences.

L186: [Mannix et al., 2018] must be replaced by the reference number.

L189:  [Truslove et al., in subm.] must be replaced by the reference number.

L230:  [Rivett et al., 2019] must be replaced by the reference number.

L234 and 235: [Back et al., 2018] and [Rivett et al., 2018] must be replaced by the reference number. The second reference has been used before, please, try to search similar references before to duplicate the same reference in different parts of the text,

L260: Table 1: in fact the authors are comparing MDG7 and SDG6. According to thihs, please specify this comparision from the beginning instead of talking about MDGs (in general terms) and SDG6 (in specific terms). Apply this comparision in the Abstract and the Introduction section.

L289: Figure 5. The source of this graphic should be provided.

L305-306: [Back et al., 2018] must be replaced by the reference number. This reference was used before.

L383: [Mannix et al., 2018] and [Rivett et al., 2019] must be replaced by the reference number. Both references were used before.

L385: [Truslave et al., 2018] must be replaced by the reference number. This reference was used before.

L423: A discussion section must be included in order to put in balance the obtained results and the previous literature.

L424: More than a Conclusion section it seems a recommendation section.

Author Response

The paper aims to provide an in-depth 20 determination of the impact of Stranded Assets for rural water supply despite international development targets, from MDGs to SDGs. In addition to this theoretical approach, a case study from Malawi is provided. The paper fits well with the aim of the journal and is well structured. However, few but important suggestions shouls be taken into account in order to improve it:

L46-48: Is the first time in which MDGs and SDGs are cited, please provide a minimum context and aim about both concepts and approaches.

Response: Completed with additional sentence.

L53: References [Caldwell 2017] and [Roper et al., 2006] should be numbered.

Response: Done

L54-58: If the definition is presented as a citation from an author or official document as appeared

between quotation marks (""), pelase provide the source of this definition.

Response:  We have changed this to single Quotation Marks as this is our definition and therefore the use the quotation marks were used to identify as a new definition for this paper.

L62: The SDG6 is cited for the first time. Please provide the context and the definition (ensure availability and sustainable management of water and sanitation for all). 

Response: Done

L72: OECD appears for the first time, please do not use the acronym.

Response: Done

L76: NGOs appears for the first time, please do not use the acronym.

Response: Done

L85: Transform the "we recognized" into more impersonal writing.

Response: Done

L89: NGO is used without acronym although it is the second time that this concept appears.

Response: Done

L101: Stranded assets not in capital letters. Apply to all the text of the paper.

Response: All done

L124-125: Figure 1. The Legend is not clearly comprehensible in the map on the left. The location map on the right is too big. Please, reorganize both maps in order to clarify the location map of the stranded assets.

Response: Done

L130 and L145: [Miller et al., 2018] must be replaced by the reference number.

Response: Done

L155: "Figure 1" must be renumbered as Figure 2.

Response: Done

L164-179: Delete "Stranded asset" from the end of all sentences.

Response:  The use of stranded asset on most of the lines clearly indicates that many of the problems resulted in stranded assets and confirms the importance of the use of this concept.

L186: [Mannix et al., 2018] must be replaced by the reference number.

Response: Done

L189:  [Truslove et al., in subm.] must be replaced by the reference number.

Response: Done

L230:  [Rivett et al., 2019] must be replaced by the reference number.

Response: Done

L234 and 235: [Back et al., 2018] and [Rivett et al., 2018] must be replaced by the reference number. The second reference has been used before, please, try to search similar references before to duplicate the same reference in different parts of the text,

Response: Done

L260: Table 1: in fact the authors are comparing MDG7 and SDG6. According to thihs, please specify this comparision from the beginning instead of talking about MDGs (in general terms) and SDG6 (in specific terms). Apply this comparision in the Abstract and the Introduction section.

Response: The Millennium Development Goals and the Sustainable Development Goals are not isolated from each other and the generalised focus of a sustainable planet is central to both efforts.   The ‘numbering’ is set by the UN and other agencies to allow for ‘Indictors’ or ‘Metrics’ or ‘KPI’s to be monitored and tracked.   Therefore in the introduction the concept of the general MDG’s and SDG’s is presented, then ‘un packed’ in the further discussion as it shows the challenge of household Metrics that are required to show progress against the ‘Indicators / KPI’s’   We therefore think it wise to generalise in the introduction and become more specific later in the paper.

L289: Figure 5. The source of this graphic should be provided.

Response: Done

L305-306: [Back et al., 2018] must be replaced by the reference number. This reference was used before.

Response: Done

L383: [Mannix et al., 2018] and [Rivett et al., 2019] must be replaced by the reference number. Both references were used before.

Response: Done

L385: [Truslave et al., 2018] must be replaced by the reference number. This reference was used before.

Response: Done

L423: A discussion section must be included in order to put in balance the obtained results and the previous literature.

Response: The Results and Discussion were combined so that the results were put into context against relevant literature and previous work as the paper progressed.   If one were to separate results from the discussions there would be considerable duplication and redundancy, and ultimately confusion to the reader.  The instructions to authors allows the Discussion with the Results and we therefore have integrated according to the instructions to authors (below).

Results: Provide      a concise and precise description of the experimental results, their      interpretation as well as the experimental conclusions that can be drawn.

Discussion: Authors      should discuss the results and how they can be interpreted in perspective      of previous studies and of the working hypotheses. The findings and their      implications should be discussed in the broadest context possible and      limitations of the work highlighted. Future research directions may also      be mentioned. This section may be combined with Results.

Conclusions: This      section is not mandatory, but can be added to the manuscript if the      discussion is unusually long or complex.

L424: More than a Conclusion section it seems a recommendation section.

Response: Agree we renamed to Recommendations